# Hydrogen Peroxide Promotes Tomato Leaf Senescence by Regulating Antioxidant System and Hydrogen Sulfide Metabolism

**DOI:** 10.3390/plants13040475

**Published:** 2024-02-07

**Authors:** Yue Yu, Siyue Wang, Wentong Guo, Meihui Geng, Ying Sun, Wanjie Li, Gaifang Yao, Danfeng Zhang, Hua Zhang, Kangdi Hu

**Affiliations:** 1School of Food and Biological Engineering, Hefei University of Technology, Hefei 230009, China; yyyu1207@163.com (Y.Y.); m18705651465_2@163.com (S.W.); zhangdanfeng@hfut.edu.cn (D.Z.); 2Key Laboratory of Cell Proliferation and Regulation Biology, Ministry of Education, College of Life Science, Beijing Normal University, Beijing 100875, China; lwj@bnu.edu.cn

**Keywords:** tomato, hydrogen peroxide, leaf senescence, antioxidant system, hydrogen sulfide

## Abstract

Hydrogen peroxide (H_2_O_2_) is relatively stable among ROS (reactive oxygen species) and could act as a signal in plant cells. In the present work, detached tomato leaves were treated with exogenous H_2_O_2_ at 10 mmol/L for 8 h to study the mechanism of how H_2_O_2_ regulates leaf senescence. The data indicated that H_2_O_2_ treatment significantly accelerated the degradation of chlorophyll and led to the upregulation of the expression of leaf senescence-related genes (*NYC1*, *PAO*, *PPH*, *SGR1*, *SAG12* and *SAG15*) during leaf senescence. H_2_O_2_ treatment also induced the accumulation of H_2_O_2_ and malondialdehyde (MDA), decreased POD and SOD enzyme activities and inhibited H_2_S production by reducing the expression of *LCD1/2* and *DCD1/2.* A correlation analysis indicated that H_2_O_2_ was significantly and negatively correlated with chlorophyll, the expression of leaf senescence−related genes, and *LCD1/2* and *DCD1/2*. The principal component analysis (PCA) results show that H_2_S showed the highest load value followed by O_2_^•−^, H_2_O_2_, *DCD1*, *SAG15*, etc. Therefore, these findings provide a basis for studying the role of H_2_O_2_ in regulating detached tomato leaf senescence and demonstrated that H_2_O_2_ plays a positive role in the senescence of detached leaves by repressing antioxidant enzymes and H_2_S production.

## 1. Introduction

Senescence is the final stage of plant development, which includes two types of senescence: mitotic and post-mitotic senescence [1]. Leaves are organs that characterize plants as autotrophic organisms and leaf senescence is a kind of post-mitotic senescence. As leaves undergo senescence, chlorophyll degradation is initiated and leads to chloroplast degeneration. Additionally, macromolecules, including proteins, lipids and nucleic acids, are also catabolized to small molecules which are exported to other developing organs, such as new buds, flowers, fruits, etc. Chloroplasts constitute about 70% of the total proteins in leaves, and thus the components’ catabolism is critical for C/N mobilization. Chlorophyll degradation causes the first visible changes of leaf senescence. Chlorophyll breakdown is initiated from the conversion of chlorophyll b to chlorophyll a, which is catalyzed by two chlorophyll b reductases, NONYELLOW COLORING 1 (NYC1) and NYC1-like (NOL). Additionally, STAYGREEN (SGR), Pheophytin Pheophorbide Hydrolase (PPH) and Pheophorbide a Oxygenase (PAO) also participate in chlorophyll degradation [2]. Leaf senescence is a highly coordinated process regulated by hundreds of *senescence-associated genes* (*SAGs*) whose transcripts increase with leaf age [2]. Among them, *SAG12* and *SAG15* are the predominant *SAGs* that are up-regulated significantly during age-related senescence [3].

Many phytohormones function to integrate environmental or developmental signals to regulate plant senescence. Ethylene is a gaseous signal that could promote leaf senescence and fruit ripening. Additionally, abscisic acid (ABA), a plant hormone regulating plants’ responses to abiotic and biotic stresses, shows increases during leaf senescence and the exogenous application of ABA could trigger leaf senescence [4]. In addition to the traditional phytohormones, reactive oxygen species (ROS) are indispensable for plant senescence [5]. ROS are produced as byproducts of aerobic energy metabolism and the excessive accumulation of ROS can lead to oxidative stress and, consequently, to damage to macromolecules and membranes [6]. The ROS, including singlet oxygen (^1^O_2_), superoxide radical (O_2_^•−^), hydrogen peroxide (H_2_O_2_), etc., are extremely reactive and H_2_O_2_ can efficiently oxidize the active thiols and travel long distances by passing through the biological membranes via aquaporins; since H_2_O_2_ is more stable compared to other ROS and easily diffuses across membranes between different cellular compartments, it could act as a signaling molecule to transmit external signals to intracellular pathways [7]. H_2_O_2_ plays an important role during plant senescence, and it could be used as a signal to promote senescence in different plant species [8,9]. At high concentrations, it could lead to cell death during the final stages of senescence. Chloroplasts, an important source of ROS in photosynthetic plant cells, are also targets of ROS-induced damages [10]. H_2_O_2_ application was found to induce the expression of the NAC transcription factor *ORS1*, which triggers the expression of senescence-associated genes and accelerates senescence [11]. However, whether and how H_2_O_2_ application affects the senescence of detached tomato leaves is still unclear.

Hydrogen sulfide (H_2_S) exposure at high concentrations is highly toxic to all animals. However, H_2_S was found to be produced endogenously in animal cells and plants [12]. In recent decades, H_2_S was found to regulate multiple aspects of plant growth and development, such as seed germination, root formation, stomatal movement, etc. [13,14]. Additionally, H_2_S could effectively delay fruit ripening and senescence by repressing ethylene synthesis and signaling pathways in strawberries, tomatoes, pears, etc. [15,16,17]. Moreover, H_2_S could activate the activities of antioxidant enzymes such as catalase (CAT), superoxide dismutase (SOD) and peroxidase (POD), which may help to alleviate ROS stress during fruit ripening and senescence [16]. However, there is no clue whether H_2_O_2_ interferes with the generation of H_2_S. Thus, in the present work, H_2_O_2_ was applied to detached tomato leaves and its effects on leaf senescence and H_2_S metabolism were explored.

## 2. Results

### 2.1. Effect of H_2_O_2_ Treatment on Detached Tomato Leaf Senescence

H_2_O_2_ is a potent alternative that is used as a sanitizing agent for horticultural products, whereas whether it could promote leaf senescence is still unclear. Thus, we treated mature tomato leaves with 10 mmol/L of H_2_O_2_ for 8 h, and subjected them to storage for another 3 days. As shown in Figure 1A, after 24 h of storage, H_2_O_2_ induced yellow and brown spots on the leaves which became more obvious compared with the control leaves on day 3. Then, the contents of chlorophyll a, b and total chlorophyll were determined. As shown in Figure 1B, chlorophyll a content was almost kept at a relative stable level till 24 h of storage, followed by a significant decrease on day 3 in both the H_2_O_2_-treated and control leaves. As shown in Figure 1C, the content of chlorophyll b decreased gradually till 24 h and dropped on day 3, suggesting that the degradation of chlorophyll b preceded that of chlorophyll a. Additionally, H_2_O_2_ treatment caused significantly lower chlorophyll b and total chlorophyll contents in contrast with the control, suggesting that H_2_O_2_ elicits tomato leaf senescence during storage. Then, the genes evolved in chlorophyll degradation, including *NYC1*, *PAO*, *PPH*, *SGR1*, *SAG12* and *SAG15*, were determined at the transcriptional level. NYC1, PAO and PPH are key chlorophyll-degrading enzymes during leaf senescence. Additionally, the SGR (STAY GREEN) protein affects chlorophyll degradation by interacting with chlorophyll-degrading enzymes, and SGR1 in tomatoes promotes chlorophyll degradation. SAGs also play an important role in leaf senescence [2]. As shown in Figure 1E, *NYC1* remained at a low level during the 8 h of H_2_O_2_ treatment and increased significantly at 24 h of storage, whereas H_2_O_2_ treatment induced significantly higher *NYC1* expression at 24 h and 3 days of storage. The expression of *PAO* was also induced at 24 h of storage, but no significant changes were observed between H_2_O- and H_2_O_2_-treated leaves. The *PPH* expression in Figure 1G indicates that *PPH* increased in expression, and H_2_O_2_ treatment induced significantly higher expression at 8 h and 3 days. Similarly, increasing trends in the expression of *SGR1*, *SAG12* and *SAG15* were also observed during leaf storage. H_2_O_2_ could induce higher expression in *PPH* at 8 h, 24 h and 3 days; in *SGR1* and *SAG12* at 24 h and 3 days; and in *SAG15* at 24 h. In sum, the above data suggest that H_2_O_2_ could act as a signal to elicit leaf senescence during storage by accelerating chlorophyll degradation and senescence-related gene expressions.

### 2.2. Effect of H_2_O_2_ Treatment on Reactive Oxygen Species in Tomato Leaves during Storage

ROS have been demonstrated as critical initiators of plant senescence. Thus, we determined the content of ROS-related metabolites in tomato leaves during storage. As shown in Figure 2A, the content of H_2_O_2_ increased gradually during the storage of leaves, and the content in H_2_O_2_-treated tomato leaves was significantly higher than that in the control. Figure 2B shows the changes in O_2_^•−^ in tomato leaves during storage, and H_2_O_2_ treatment was able to induce a significantly higher level of O_2_^•−^ at 8 h and 24 h of storage. Similar to the change pattern of H_2_O_2_, the content of MDA increased during leaf storage and H_2_O_2_ treatment caused a significantly higher accumulation of MDA in tomato leaves during 3 days of storage. To obtain visible evidence of ROS metabolism in tomato leaves, Figure 2D shows the phenotype of tomato leaves on day 3 stained by DAB which indicate the content of H_2_O_2_. It was found that H_2_O_2_ treatment caused a deeper level of browning, suggesting that exogenous H_2_O_2_ treatment induced the accumulation of H_2_O_2_ in tomato leaves. Additionally, the dead cells in tomato leaves after storage for 3 days were visualized by trypan blue staining. And the image in Figure 2E indicates that more cells underwent cell necrosis during storage. These results suggest that H_2_O_2_ caused an imbalanced ROS metabolism in leaves, which may contribute to accelerated leaf senescence in tomato leaves during storage.

### 2.3. Effect of H_2_O_2_ Treatment on the Activities of Antioxidant Enzymes

To further investigate the changes in ROS metabolism under H_2_O_2_ treatment, the enzymes responsible for ROS metabolism were studied at the activity level. The first line of defense against ROS in plants is SOD, which converts O_2_^•−^ to H_2_O_2_. In addition, POD and CAT are actively involved in the decomposition of H_2_O_2_ into H_2_O and O_2_^•−^ [18]. As shown in Figure 3A, the activity of CAT increased gradually during leaf storage and no significant difference was found between the control and H_2_O_2_ treatment. Figure 3B shows the changes in POD activity during leaf senescence; it was found that POD exhibited a decreasing trend and H_2_O_2_ treatment caused significantly lower POD activity at 24 h and 3 days of storage. Figure 3C shows that the SOD activity increased slightly at 8 h in the control leaves, whereas the increase was attenuated in H_2_O_2_-treated leaves followed by a decrease in both the control and H_2_O_2_ treatment. Generally, H_2_O_2_ caused the decreased activity of POD and SOD during tomato leaf senescence, which may contribute to earlier senescence in H_2_O_2_-treated leaves.

### 2.4. Effect of H_2_O_2_ on Hydrogen Sulfide Metabolism in Tomato Leaves

Previous research suggests that H_2_S could alleviate postharvest senescence in horticultural products by repressing the accumulation of ROS [16]. However, whether H_2_O_2_ could modulate H_2_S metabolism is still not clear. Thus, we evaluated the changes in the content of H_2_S in H_2_O_2_-treated tomato leaves during storage. Figure 4A shows the results of H_2_S production evaluated by lead acetate H_2_S detection strips, and the results showed that H_2_O_2_ treatment caused lower H_2_S contents especially at 8 h of H_2_O_2_ treatment. Furthermore, Figure 4B shows that H_2_O_2_ treatment resulted in a significantly lower rate of H_2_S production compared to the control. DCD is a key enzyme in the production of H_2_S using D-Cys as the substrate, while LCD catalyzes the decomposition of L-Cys to H_2_S, ammonia (NH_4_^+^) and pyruvate [19,20]. Then, we determined the expression of H_2_S-related metabolism genes, including *LCD1*, *LCD2*, *DCD1* and *DCD2*. As shown in Figure 4C, the expression of *LCD1* and *LCD2* showed a decreasing trend during leaf senescence and H_2_O_2_ treatment caused significantly lower expression at 3 days of storage. Additionally, the expression of *DCD1* and *DCD2*, which catalyze the production of H_2_S with D-cysteine as the substrate, were also determined. As shown in Figure 4E, F, a similar decreasing trend in the expression of *DCD1* and *DCD2* was observed, and H_2_O_2_ was found to reduce *DCD1* expression at 3 days and *DCD2* expression at 24 h and 3 days. Thus, we concluded that H_2_O_2_ could repress the generation of H_2_S by reducing the expression of *LCD1/2* and *DCD1/2*.

### 2.5. Correlation and Principal Component Analysis of Physiological Indexes and Senescence-Related Gene Expression

The correlation among the contents of H_2_O_2_ and MDA, H_2_S and the O_2_^•−^ production rate; the contents of total chlorophyll, chlorophyll a and chlorophyll b; the gene expression of *NYC1*, *PAO*, *PPH*, *SGR1*, *SAG12*, *SAG15*, *LCD1*, *LCD2*, *DCD1* and *DCD2*; and the activities of POD, SOD and CAT were analyzed. Figure 5A shows that the content of H_2_O_2_ showed higher negative correlations with total chlorophyll, chlorophyll a and chlorophyll b with correlation values of −0.909, −0.866 and −0.940, respectively. Additionally, total chlorophyll, chlorophyll a and chlorophyll b also showed negative correlations with the gene expression of *NYC1*, *PAO*, *PPH*, *SGR1*, *SAG12* and *SAG15*, while they showed positive correlations with the gene expression of *LCD1/2* and *DCD1*/*2*. H_2_S showed positive correlations with total chlorophyll, chlorophyll a, chlorophyll b, suggesting the role of H_2_S in alleviating chlorophyll degradation during leaf senescence. In all, the correlation data suggest that H_2_O_2_ may accelerate chlorophyll degradation, while H_2_S showed a protective role.

Principal component analysis (PCA) was carried out based on the data of H_2_O_2_; MDA and H_2_S content; the O_2_^•−^ production rate; total chlorophyll; chlorophyll a; chlorophyll b; the gene expression of *NYC1*, *PAO*, *PPH*, *SGR1*, *SAG12*, *SAG15*, *LCD1*, *LCD2*, *DCD1* and *DCD2*; and the activity of POD, SOD and CAT. As shown in Figure 5B, PC1 and 2 contributed to 78.7% and 18.6% of the variability in the data, respectively. The data indicated that H_2_S showed the highest load value followed by O_2_^•−^, H_2_O_2_, *DCD1*, *SAG15*, etc. Therefore, it could be concluded that H_2_S and ROS metabolites contribute to the difference in leaf senescence caused by H_2_O_2_ treatment.

## 3. Discussion

During O_2_ metabolism in aerobic organisms, a variety of ROS are produced along with cell respiration, and the incomplete reduction of O_2_ leads to the generation of O_2_^•−^, hydroxyl radical (OH^•^) and H_2_O_2_. Excessive ROS are capable of inducing cellular damage by the oxidation of proteins, membranes and the mutation of DNA sequences [5,21]. However, appropriate concentrations of ROS could act as signaling molecules [22]. Among the ROS, H_2_O_2_ has been shown to regulate plant growth, development and stress resistance because it can react with the thiol (-SH) group on cysteine residues in proteins, which can regulate the function of target proteins [23].

H_2_O_2_ was increased during fruit senescence, suggesting that endogenous H_2_O_2_ showed a role in promoting fruit senescence [24]. To investigate whether exogenous H_2_O_2_ regulates tomato leaf senescence, we treated detached tomato leaves with 10 mmol/L of H_2_O_2_ for 8 h, and subjected them to storage for 3 days. The data indicated that H_2_O_2_ treatment caused significantly lower chlorophyll contents in contrast with the control, accompanied with the increased gene expression of chlorophyll degradation genes, including *NYC1*, *PAO*, *PPH* and *SGR1*, and senescence-related genes such as *SAG12* and *SAG15*. The increased gene expression of chlorophyll degradation genes may contribute to lower chlorophyll contents in H_2_O_2_-treated leaves. *SAG12* and *SAG15* may function together to decompose proteins in leaves during senescence. Exogenous H_2_O_2_ also causes increases in H_2_O_2_, O_2_^•−^ and MDA, which is a byproduct of lipid peroxidation, and thus the disturbed ROS homeostasis may lead to ROS stress and, finally, leaf senescence. Similarly, we previously reported that the depletion of *DCD2*, which encodes a D-cysteine desulfhydrase, causes decreased H_2_S release and increased H_2_O_2_ and MDA accumulation in fruits, which leads to an imbalance in ROS metabolism [24]. It has been shown that ROS production is closely related to plant senescence metabolism, and that excessive H_2_O_2_ accumulation is an important promoter of leaf senescence [25]. H_2_O_2_, due its role of deactivating microbes, has been applied to postharvest horticultural products. For instance, H_2_O_2_ treatment on harvest longan fruit causes a higher browning index and an increased rate of O_2_^•−^ in the pericarp due to the reducing capacity of active oxygen scavenging [26]. Additionally, H_2_O_2_ treatment accelerated the accumulation of endogenous H_2_O_2_ by activating NADPH oxidase in bamboo shoots and up-regulated DNase, RNase and caspase 3-like activities, leading to the acceleration of the programmed cell death process [27]. It has also been shown that most senescence-related and ROS scavenging genes are up-regulated in the *ls1* mutant, that *LS1* may regulate leaf development and function, and that the disruption of *LS1* function promotes ROS accumulation and accelerates leaf senescence and cell death in rice [28]. Similarly, we also observed increased cell death in tomato leaves treated with H_2_O_2_ (Figure 2E). The balance between SOD and different H_2_O_2_ scavenging enzymes such as POD and CAT in the cell is thought to be critical in determining the level of ROS homeostasis [29]. To further investigate the increase in ROS metabolism under H_2_O_2_ treatment, the enzymes responsible for ROS metabolism were studied and H_2_O_2_ was found to decrease the activity of POD and SOD during tomato leaf senescence, which may contribute to earlier senescence in H_2_O_2_-treated leaves. Therefore, H_2_O_2_ could accelerate leaf senescence by increasing endogenous ROS production and promoting chlorophyll degradation through up-regulating chlorophyll decomposing genes.

H_2_S could effectively delay fruit ripening and senescence by repressing ethylene synthesis and signaling pathways or by activating the activities of antioxidant enzymes such as CAT, SOD and POD, which may help to alleviate ROS stress during fruit ripening and senescence [16]. Additionally, compared with wild-type fruits, the mutation of D-cysteine desulfhydrase *SlDCD2* induced H_2_O_2_ and MDA accumulation in fruits, which led to an imbalance in ROS metabolism. All the above evidence suggests that H_2_S could alleviate plant senescence by repressing ROS accumulation. However, there is no clue whether H_2_O_2_ interferes with the generation of H_2_S. We hypothesis that H_2_O_2_ may attenuate the production of H_2_S. To test this hypothesis, endogenous H_2_S production and the gene expression of H_2_S-releasing enzymes were analyzed. The data indicate that H_2_O_2_-treatment caused lower H_2_S contents, especially at 8 h of H_2_O_2_ treatment on detached tomato leaves. LCD (L-cysteine desulfhydrase, with L-Cys as the substrate) has also been shown to catalyze the degradation of cysteine to H_2_S, ammonia and pyruvate, and DCD uses D-cysteine as the substrate [30]. In addition, *LCD1* mutation accelerated leaf senescence, whereas *LCD1* overexpression significantly delayed leaf senescence, which may be due to endogenous H_2_S caused by *LCD1* mutation [31]. The gene expression data indicate that H_2_O_2_ could repress the generation of H_2_S by reducing the expression of *LCD1/2* and *DCD1/2*, which may contribute to decreased endogenous H_2_S in H_2_O_2_-treated leaves.

To study the relations between H_2_O_2_ and other parameters, the correlation among H_2_O_2_, MDA and H_2_S content; the O_2_^•−^ production rate; total chlorophyll; chlorophyll a; chlorophyll b; the gene expression of *NYC1*, *PAO*, *PPH*, *SGR1*, *SAG12*, *SAG15*, *LCD1*, *LCD2*, *DCD1* and *DCD2*; and the activity of POD, SOD and CAT were analyzed. The data revealed that H_2_O_2_ showed higher negative correlations with total chlorophyll, chlorophyll a and chlorophyll b. Additionally, chlorophyll also showed negative correlations with the gene expression of *NYC1*, *PAO*, *PPH*, *SGR1*, *SAG12* and *SAG15*, whereas it showed positive correlations with the gene expression of *LCD1*, *LCD2*, *DCD1* and *DCD2*. H_2_S showed positive correlations with total chlorophyll, chlorophyll a and chlorophyll b, suggesting the role of H_2_S in alleviating chlorophyll degradation during leaf senescence, but a negative correlation with H_2_O_2_. In all, the correlation data suggest that H_2_O_2_ may accelerate chlorophyll degradation, while H_2_S showed a protective role, and there is an antagonizing relation between H_2_O_2_ and H_2_S.

In summary, the role of H_2_O_2_ in regulating detached tomato leaf senescence was explored. Exogenous H_2_O_2_ application could promote chlorophyll degradation by increasing the expression of chlorophyll degradation-related genes. Additionally, H_2_O_2_ causes excessive ROS accumulation through repressing the activities of the antioxidant enzymes POD and SOD. By analyzing endogenous H_2_S content and the expression of *LCDs/DCDs*, H_2_S production was attenuated by H_2_O_2_ treatment, which may contribute to accelerated leaf senescence, suggesting the antagonizing relation between H_2_O_2_ and H_2_S. Thus, we provide solid evidence that H_2_O_2_ plays a positive role in detached leaf senescence, and H_2_O_2_ exhibits an antagonizing relation with H_2_S.

Controlling senescence can greatly improve crop yields and other plant characteristics, such as extending shelf life, especially in light of potential future food shortages and the use of plants as bioenergy sources [32]. Understanding how leaf senescence is regulated will help control senescence in the future, either through genetic modification or the manipulation of key environmental triggers [33]. In the present work, we found that H_2_O_2_ could act as a signal to promote leaf senescence by inhibiting H_2_S production. Thus, genetic manipulation on H_2_O_2_ metabolism-related genes such SOD, CAT and POD may provide a valuable tool to delay leaf senescence in crops or horticulture plants, thereby improving the yield.

## 4. Materials and Methods

### 4.1. Plant Material and Growing Conditions

The wild-type tomatoes (*Solanum lycopersicum* cv. “Micro-Tom”) used in this study were grown in a growth chamber (23 ± 2 °C; 50–70% relative humidity) under 16 h of light/8 h of darkness. Plants were grown to 6 weeks of age and tomato leaves in the same leaf position were removed for subsequent treatments.

### 4.2. Hydrogen Peroxide Treatment of Tomato Leaves

According to a previous study [34], we soaked tomato leaves in the same leaf position and growth period using 10 mmol/L of H_2_O_2_ for 8 h, and set the H_2_O treatment as a control. For the selection of the appropriate concentration of H_2_O_2_ and the treatment time, tomato leaves were treated with H_2_O_2_ at concentrations of 500 µmol/L, 1 mmol/L, 10 mmol/L and 100 mmol/L; the leaf phenotypes were recorded for 5 days; and thus the treatment time of 8 h and 10 mmol/L were selected as the treatment procedure that promoted leaf senescence obviously. Samples collected after 0 h, 8 h, 24 h and 3 d of treatment were used for mRNA extraction and qPCR.

### 4.3. RNA Extraction and RT–qPCR

RNA was extracted from a 0.5 g leaf and the first strand cDNA was synthesized following the method reported previously [31]. Tomato *Tubulin* was used as an internal reference.

### 4.4. Determination of the Levels of Chlorophyll in Tomato Leaves

The tomato fruit samples without seeds were uniformly ground into powder under freezing conditions, mixed with 5 mL of acetone and hexane (2:3), centrifuged at 4000 rpm for 5 min to collect the supernatant, and the absorbance values were determined by an enzyme meter at 663 nm, 645 nm, 505 nm and 453 nm. The contents of chlorophyll a, b and total chlorophyll were calculated by the formula described by Nagata et al. [35].

### 4.5. MDA Content

Based on the methodology reported by Li [36], 0.5 g of leaf samples was homogenized with 5% trichloroacetic acid, incubated and then centrifuged to collect the supernatant. The absorbance was measured at 450, 532 and 600 nm.

### 4.6. H_2_O_2_ Content

For the determination of the H_2_O_2_ content, 0.5 g of tomato leaves was taken, homogenized in 3 mL of pre-cooled acetone, centrifuged at 12,000 rpm for 30 min and the absorbance value at 508 nm was measured to determine the H_2_O_2_ content [37].

### 4.7. Production Rate of O_2_^•−^

As mentioned previously [38], the reaction buffer was composed of 50 mM phosphate buffer (pH 7.8) containing 17 mM sulfanilic acid, 1 mM hydroxylamine hydrochloride, 7 mM 1-naphthylamine and 50 µL sample solution. The absorbance of the mixture was measured at 530 nm, and the production rate of O_2_^•−^ was calculated using previously described formulas.

### 4.8. Antioxidant Enzyme Assay

Tomato tissue (0.5 g) was extracted using 10 mL of 50 mM phosphate buffer (pH 7.8) at 4 °C. Then, samples were centrifuged at 10,000× *g* and 4 °C for 15 min. The supernatant was the crude antioxidant enzyme solution. CAT, SOD and POD activities were measured and calculated as described previously [39,40,41]. An increase in absorbance of 1.0 × 10^−5^ OD_470_ nm·min^−1^ was considered 1 U of POD activity, a decrease in absorbance of 1.0 × 10^−3^ at OD_240_ nm·min^−1^ was considered 1 U of CAT activity and the amount used to inhibit 5% of the photochemical reduction of NBT was considered 1 U of SOD activity. The results are expressed on an FW (Fresh Weight) basis as U·g^−1^.

### 4.9. H_2_S Production in Tomato Leaves

H_2_S production was determined using the lead sulfide method [42]. The leaf samples were ground to a powder in liquid nitrogen, homogenized in 5 mL of buffer (containing 100 mM potassium dihydrogen phosphate buffer pH 7.4, 10 mM Cys and 2 mM pyridoxine 5′-phosphate) and centrifuged to obtain a supernatant. Five sheets of lead acetate H_2_S test paper were placed in an airtight container above the liquid phase and incubated at 37 °C, protected from light, for several hours until the paper darkened. We soaked the test paper with 1 mL of 1 M HCl and it further reacted with N, Ndimethyl-phenylenediamine (DPD) with FeCl_3_, which was detected colorimetrically at 670 nm.

### 4.10. Statistical Analysis

The data were analyzed by SPSS statistics 25 software (SPSS version 25.0; Armonk, NY, USA), and the experimental data were plotted and analyzed by Excel software (Microsoft Office 2016). The correlation and the principal component analysis (PCA) of the data above were analyzed by the tools on the OmicShare (omicshare.com, accessed on 16 September 2023) platform.

## Figures and Tables

**Figure 1 plants-13-00475-f001:**
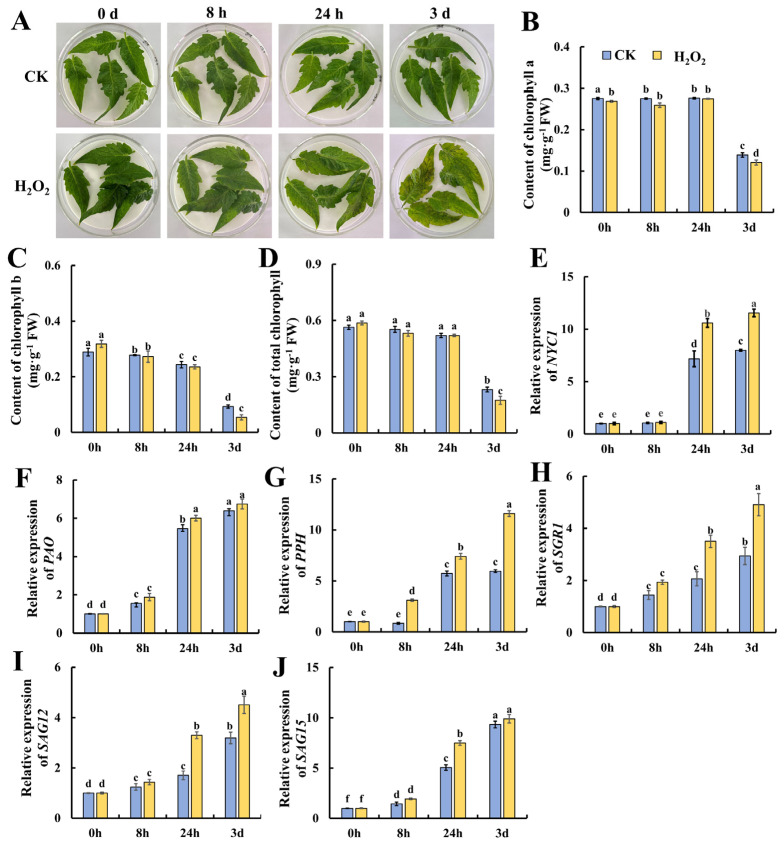
Accelerated senescence of tomato leaves by hydrogen peroxide treatment leads to reduced chlorophyll levels and the up-regulation of the expression of genes related to chlorophyll degradation. (**A**) Phenotypic changes in tomato leaves at 0 h, 8 h, 24 h and 3 d after treatment. Changes in (**B**) chlorophyll a, (**C**) chlorophyll b and (**D**) chlorophyll content, and the gene expressions of (**E**) *NYC1*, (**F**) *PAO*, (**G**) *PPH*, (**H**) *SGR1*, (**I**) *SAG12* and (**J**) *SAG15* at different treatment times and in different treatment groups. Data are expressed as mean ± SD (n = 3), and significant differences at the level of *p* < 0.05 are indicated with different letters.

**Figure 2 plants-13-00475-f002:**
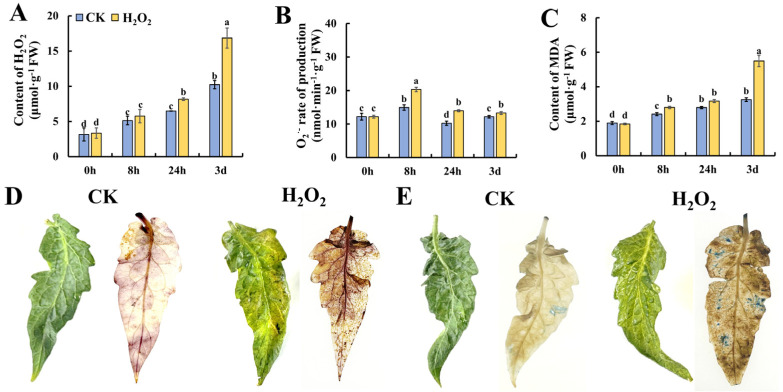
Effect of H_2_O_2_ treatment on ROS homeostasis in tomato leaves. (**A**–**C**) Content of H_2_O_2_ (**A**), production rate of O_2_^•−^ (**B**) and content of MDA (**C**) in tomato leaves. DAB (**D**) and trypan blue (**E**) staining of leaves on day 3 after H_2_O_2_ treatment. Data are expressed as mean ± SD (n = 3), and significant differences at the level of *p* < 0.05 are indicated with different letters.

**Figure 3 plants-13-00475-f003:**
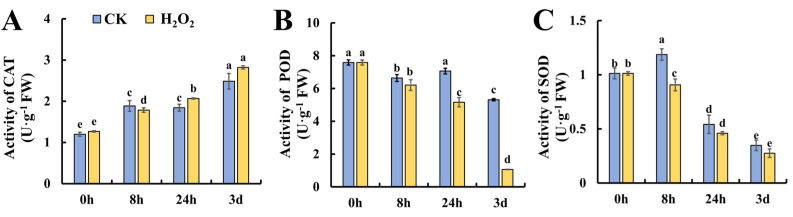
Effect of H_2_O_2_ treatment on antioxidant enzyme activities in tomato leaves. (**A**–**C**) Activities of CAT (**A**), POD (**B**) and SOD (**C**) in tomato leaves under H_2_O_2_ and H_2_O treatment. Data are expressed as mean ± SD (n = 3), and significant differences at the level of *p* < 0.05 are indicated with different letters.

**Figure 4 plants-13-00475-f004:**
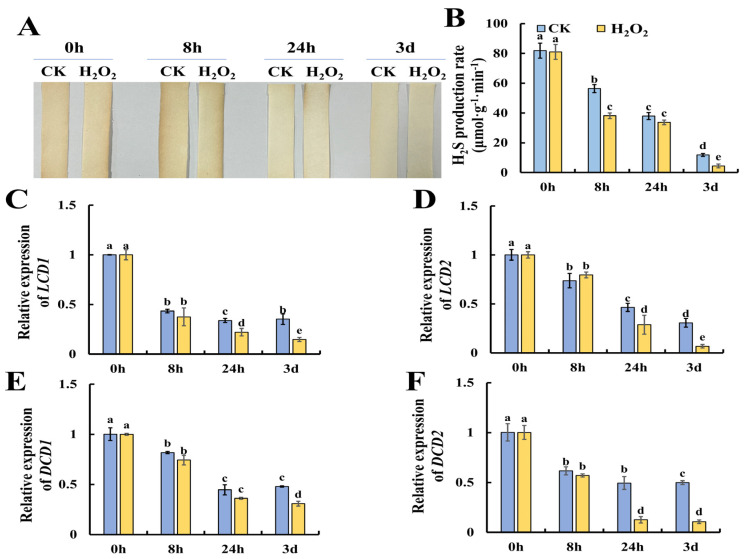
Reduction in endogenous hydrogen sulfide production in tomato leaves by H_2_O_2_ treatment. (**A**,**B**) Tomato leaves with cysteine as substrates were assayed by lead acetate test paper and a hydrochloric acid absorption well method. (**C**–**F**) Expression patterns of *LCD1* (**C**), *LCD2* (**D**), *DCD1* (**E**) and *DCD2* (**F**) in tomato leaves under H_2_O_2_ and H_2_O treatment. Data are expressed as mean ± SD (n = 3), and significant differences at the level of *p* < 0.05 are indicated with different letters.

**Figure 5 plants-13-00475-f005:**
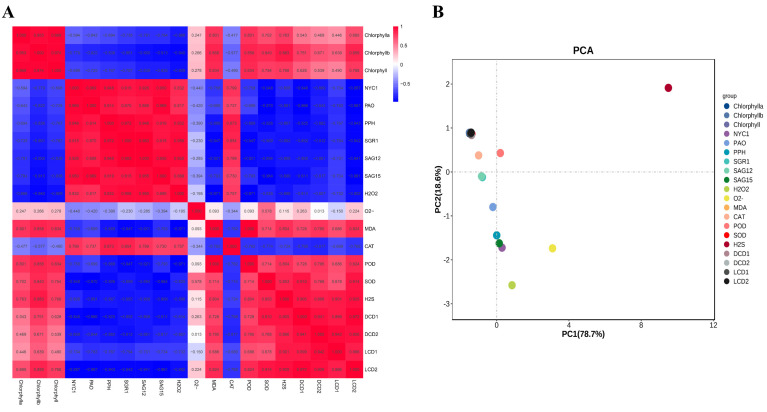
(**A**) Correlation analysis among the parameters of chlorophyll a, chlorophyll b, chlorophyll, H_2_O_2_, MDA, the production of O_2_^•−^ and H_2_S, and the gene expressions of *NYC1*, *PAO*, *PPH*, *SGR1*, *SAG12*, *SAG15*, *DCD1*, *DCD2*, *LCD1* and *LCD2* in tomato leaves at 0, 8, 24 h and 3 d after H_2_O_2_ and H_2_O treatments. The correlation indexes are as follows: “+” indicates a positive correlation, “−” indicates a negative correlation, 0.8–1 indicates a high correlation, 0.6–0.8 indicates a strong correlation, 0.4–0.6 indicates a moderate correlation and 0.2–0.4 indicates a moderate correlation. (**B**) PCA of the testing indicators in tomato leaves at 0, 8, 24 h and 3 d after H_2_O_2_ and H_2_O treatments. PC1 and PC2, respectively, represent the contribution rate of principal components.

## Data Availability

Data are contained within the article.

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
