# Peer review of "Hydrogen Peroxide Promotes Tomato Leaf Senescence by Regulating Antioxidant System and Hydrogen Sulfide Metabolism"

_plants, 2024, doi:10.3390/plants13040475_

Round 1

Reviewer 1 Report

Comments and Suggestions for Authors

The manuscript by Yue et al. investigates the role of hydrogen peroxide (H2O2) in tomato leaf senescence. The findings are novel and therefore extend our knowledge of H2O2 in leaf senescence process. There are some concerns that should be addressed.

Major concerns

The statistics of most data should be improved. Take Fig 1C for example, the corresponding context is “As shown in figure 1C, the content of chlorophyll b decreased gradually till 24 h and dropped on day 3”, however, there’s no statistics to support this claim. The one-way ANOVA statistics, if possible, could be performed followed by using different letters to represent significant differences.

Minor concern

1, Line 55: “tis” might be “its”; Line 98: “induced” should be “induce”.

2, Line 99: “PPH at 8 h, 24 h and 3 days”. “24 h” has no differences.

3, Line 123: “H2O2 treatment induced the accumulation of H2O2 treated tomato leaves”. This sentence should be rephrased.

4, Regarding Figure 2D and 2E, please provide more detailed descriptions in Figure legend. It seems that before H2O2 treatments, the leaves already had certain differences.

5, Line 126: “This result suggest that H2O2 caused imbalanced ROS metabolism”. First, “This result” should be “These results”, or “suggest” change to be “suggests”. Second, please specify why H2O2 caused imbalanced ROS metabolism.

6, Line 143-144: “H2O2 caused decreased activity of POD and SOD during tomato leave senescence”. I have concerns regarding the SOD data. It seems that no differences of SOD activities were observed between CK and H2O2 treatments in 24 h and 3 days. Differences only observed at the 8 h time point seem not very much convictive to conclude that the activity of SOD is regulated by H2O2. On the other hand, how to understand that in the CK group, the activity of SOD increases first and then decreases?

7, Line 153. “ROS” might be “H2S”.

8, Line 156-158: “Further the production rate of H2S suggested that H2O2 treatment caused significantly lower level of H2S in comparison to control leaves.” This sentence may need to be rephrased.

9, Line 165-166: “we conclude that H2O2 could repress the generation of H2S by reducing the expression of LCD1/2 and DCD1/2.” How to explain that the differences of H2S production happened at 8 h while the expression level differences of these genes happened at 24 h or 3 days?

10, Please provide more information about the correlation analysis to make it easier to understand.

11, Line 194-195: “Therefore, it could be concluded that H2S and ROS metabolites contribute to the difference in leaf senescence caused by H2O2 treatment”. Please make it clear how the PCA analysis could reach this conclusion.

Others (optional)

Regarding Figure 1, a brief illustration showing the functions of the genes tested may be helpful. Similarly, in terms of Figure 3 and Figure 4, illustrations showing roles of the enzymes tested would also be helpful.

Comments on the Quality of English Language

Moderate editing of English language is needed.

Author Response

Major concerns: The statistics of most data should be improved. Take Fig 1C for example, the corresponding context is “As shown in figure 1C, the content of chlorophyll b decreased gradually till 24 h and dropped on day 3”, however, there’s no statistics to support this claim. The one-way ANOVA statistics, if possible, could be performed followed by using different letters to represent significant differences.

Response: Thank you for pointing out the shortcomings. Based on your comments, in the resubmitted manuscript, we performed a one-way ANOVA on the data and labeled significant differences in the graphs with different letters.

  1. Line 55: “tis” might be “its”; Line 98: “induced” should be “induce”.

Response: We sincerely thank the reviewer for careful reading. In our resubmitted manuscript, the typo is revised.

  1. Line 99: “PPH at 8 h, 24 h and 3 days”. “24 h” has no differences.

Response: We greatly appreciate reviewer’s nice comments. It is sorry that the difference symbol was missing in the figure but actually there is significant difference. We performed a one-way ANOVA on the data and found that the expression of PPH is indeed differed significantly at 24 h.

  1. Line 123: “H2O2 treatment induced the accumulation of H2O2 treated tomato leaves”. This sentence should be rephrased.

Response: We feel sorry for our carelessness. In our resubmitted manuscript, this sentence is rewritten as follows “Exogenous H2O2 treatment induced the accumulation of H2O2 in tomato leaves”. Please see the revision on line 129-130.

  1. Regarding Figure 2D and 2E, please provide more detailed descriptions in Figure legend. It seems that before H2O2 treatments, the leaves already had certain differences.

Response: Thanks for your suggestion. In accordance with your suggestions, we have added explanations in the legend. Because the leaves we used for staining were 3 days after the H2O2 treatment or control, there were some differences in the leaves. We use similar leaves for the treatment, so the differences between each leaf is very minor. Please see the revision on line 137-138.

  1. Line 126: “This result suggest that H2O2 caused imbalanced ROS metabolism”. First, “This result” should be “These results”, or “suggest” change to be “suggests”. Second, please specify why H2O2 caused imbalanced ROS metabolism.

Response: Thank you for valuable comment. The typo at line 126 has been corrected in accordance with the reviewer's comments.

In this study we examined the ROS content in tomato leaves of experimental and control groups and found that the experimental group exhibited higher ROS content compared to the control group. Normal physiological processes produce small amounts of ROS, which are essential for metabolism, physiological regulation. Previous studies have shown that SOD catalyzes the dismutation of O2•− to molecular oxygen and H2O2, and POD eliminates H2O2. The content of O2•− and H2O2 is controlled by different antioxidant enzyme activities. From the present study, it can be seen that external hydrogen peroxide treatment reduced the enzyme activity of POD and SOD, thereby leading to excessive accumulation of ROS.

  1. Line 143-144: “H2O2 caused decreased activity of POD and SOD during tomato leave senescence”. I have concerns regarding the SOD data. It seems that no differences of SOD activities were observed between CK and H2O2 treatments in 24 h and 3 days. Differences only observed at the 8 h time point seem not very much convictive to conclude that the activity of SOD is regulated by H2O2. On the other hand, how to understand that in the CK group, the activity of SOD increases first and then decreases?

Response: Thank you for your valuable comment. The time point of 8 h is the time duration of H2O2. The figure indicates the 8 h H2O2 treatment induces lower SOD activity. However, the difference at SOD activity between H2O2 treatment and control disappeared thereafter. In control leaves, at the beginning of leaf senescence, SOD may be activated in response to endogenous ROS accumulation, but after long term storage, the tissue of leaf may undergo protein decomposition and thus SOD activity decreased in a long term.

  1. Line 153. “ROS” might be “H2S”.

Response: Thank you for this careful comment. In our resubmitted manuscript, the typo is revised.

  1. Line 156-158: “Further the production rate of H2S suggested that H2O2 treatment caused significantly lower level of H2S in comparison to control leaves.” This sentence may need to be rephrased.

Response: We feel sorry for our carelessness. In our resubmitted manuscript, this sentence is rewritten as follows “Furthermore, Figure 4B shows that H2O2 treatment resulted in a significantly lower rate of H2S production compared to the control”. Please see the revision on line 166-167.

  1. Line 165-166: “we conclude that H2O2 could repress the generation of H2S by reducing the expression of LCD1/2 and DCD1/2.” How to explain that the differences of H2S production happened at 8 h while the expression level differences of these genes happened at 24 h or 3 days?

Response: Thank you for pointing this issue. H2O2 treatment will cause ROS-rich environment. LCD and DCD possibly undergo post-translational modification due to H2O2, which may modify LCD/DCD at protein level as many works suggest that H2O2 modifies cysteine residues in protein which caused the transition of –SH group to –SOH group. Thus, we suppose that the modification mediated by H2O2 may inhibit LCD/DCD activity directly, which causes the changes of LCD/DCD activity before the changes at expression level. The related work can be found at Bi, G., Hu, M., Fu, L. et al. The cytosolic thiol peroxidase PRXIIB is an intracellular sensor for H2O2 that regulates plant immunity through a redox relay. Nat. Plants 8, 1160–1175 (2022).

  1. Please provide more information about the correlation analysis to make it easier to understand.

Response: In response to your suggestion, we have added the meanings represented by the correlation values in the figure legend. Please see the revision on line 212-214.

  1. Line 194-195: “Therefore, it could be concluded that H2S and ROS metabolites contribute to the difference in leaf senescence caused by H2O2 treatment”. Please make it clear how the PCA analysis could reach this conclusion.

Response: Thank you for valuable comment. In order to better compare the relationship between different indicators, the physiological indicators detected by PCA analysis were used in this study. The contribution rates of the first and second principal components are 78.7% and 18.6%, and the cumulative contribution rate is 97.3%, which indicates that the first two principal components reflect most of the information of the original variables and meet the requirements of the analysis. H2S production rate, O2•− production rate and the content H2O2 were positively correlated with PC1, with H2S accounting for the highest percentage of the total. In PC2, H2S production rate, the content H2O2, O2•− production rate, and the expression of DCD1 accounted for the highest percentage. These data suggest that H2S and ROS metabolites are the main contributors to leaf senescence due to exogenous H2O2 treatment.

Others (optional)

Regarding Figure 1, a brief illustration showing the functions of the genes tested may be helpful. Similarly, in terms of Figure 3 and Figure 4, illustrations showing roles of the enzymes tested would also be helpful.

Response: Thank you for your valuable comment. According to your suggestion, we have added a description of the function of the genes tested in the text, the details as followed: NYC1, PAO and PPH are key chlorophyll-degrading enzymes during leaf senescence. Besides, SGR (STAY GREEN) protein affects chlorophyll degradation by interacting with chlorophyll-degrading enzymes, and SGR1 in tomato promotes chlorophyll degradation. SAGs also play an important role in leaf senescence [2]. Please see the revision on line 93-97. In our resubmitted manuscript, we have also added a description of the function of the relevant enzymes tested, the details as followed:1. The first line of defense against ROS in plants is SOD, which converts O2•− to H2O2. In addition, POD and CAT are actively involved in the decomposition of H2O2 into H2O and O2•− [18]. 2. DCD is a key enzyme in the production of H2S using D-Cys as the substrate, while LCD catalyzes the decomposition of L-Cys to H2S, ammonia (NH4+) and pyruvate [19,20]. Please see the revision on line 142-145 and 167-169.

Reviewer 2 Report

Comments and Suggestions for Authors

In this study, the authors investigate the hydrogen peroxide promotes tomato leaf senescence by the detached tomato leaf, and find H2O2 plays a positive role in the senescence of isolated leaves and that H2O2 is antagonistic to H2S.

 Minor comments:

 Line 18, there should be a .after DCD1/2.

Line 40-41 is not clear.

Line 55 is not clear.

In the Materials and Methods part, how did the authors grow the tomato plants? In the field or greenhouse? How many plants were used? How long did the authors grow the plants and then cut the leaves? Need to give more details.

The authors mentioned that phytohormones were involved in plant senescence in the Introduction and Discussion, but did not measure the phytohormones in this experiment.

The format of the subtitle should be consistent. 

Author Response

Minor comments:

  1. Line 18, there should be a “.” after DCD1/2.

Response: We sincerely thank you for the careful reading. We added a “.” in line 18.

  1. Line 40-41 is not clear.

Response: We apologize for the clarity of our presentation. The unclearly expressed sentence in line 40-42 was changed as followed:Leaf senescence is a highly coordinated process regulated by hundreds of senescence-associated genes (SAGs) whose transcripts increase with leaf age”.

  1. Line 55 is not clear.

Response: We apologize for the clarity of our presentation. The unclearly expressed sentence in line 55 was changed as followed: “Since H2O2 is more stable compared to other ROS and easily diffuses across membranes between different cellular compartments”.

  1. In the Materials and Methods part, how did the authors grow the tomato plants? In the field or greenhouse? How many plants were used? How long did the authors grow the plants and then cut the leaves? Need to give more details.

Response: Thank you for valuable comment. Based on your suggestions, we have added to the Material and Methods section as follows: Wild-type tomatoes (Solanum lycopersicum cv. “Micro-Tom”) used in this study were grown in a growth chamber (23 ± 2°C; 50-70% relative humidity) under 16 h of light/8 h of darkness. Plants were grown to 6 weeks of age and mature tomato leaves from six plants at the same leaf position were removed for each treatment. Please see the revision on line 311-315.

  1. The authors mentioned that phytohormones were involved in plant senescence in the Introduction and Discussion, but did not measure the phytohormones in this experiment.

Response: Thank you for pointing out the shortcomings of this paper. Although phytohormones are also key factors in the regulation of leaf senescence, it is the antioxidant system and the regulatory relationship with hydrogen sulfide that are the focus of attention among us. We will look more deeply into the effects of oxidative stress on plant hormone levels in future studies.

  1. The format of the subtitle should be consistent.

Response: Thank you for this careful comment. Based on your suggestion, we have formatted the subheadings to be consistent.

Reviewer 3 Report

Comments and Suggestions for Authors

This study examines various aspects, including the effects of H2O2 treatment on reactive oxygen species, antioxidant enzymes, and hydrogen sulfide metabolism in tomato leaves. Followings are some comments to the study: (1) The experimental design is adequate, but there's a lack of detail in describing the methods, particularly in the selection of concentrations and the rationale behind the duration of treatments. A more thorough description would enhance the reproducibility of the study. (2) The connection between the observed effects and the underlying biochemical pathways could be explored more thoroughly. Include a discussion on how these findings align or contrast with existing literature. (3) The conclusion appropriately summarizes the findings but lacks depth in terms of future research directions. Elaborating on how this research could pave the way for new studies or applications in agriculture would be valuable. (4) In the References section, some recent studies could be added to provide a more current context to the research. Ensure that all references are up to date and relevant.

Comments on the Quality of English Language

Moderate editing of English language required

Author Response

  1. The experimental design is adequate, but there's a lack of detail in describing the methods, particularly in the selection of concentrations and the rationale behind the duration of treatments. A more thorough description would enhance the reproducibility of the study.

Response: We sincerely thank you for your constructive suggestions on the MS. For the selection of appropriate concentration of H2O2 and treatment time, tomato leaves were treated with H2O2 at concentrations of 500 µmol/L, 1 mmol/L, 10 mmol/L, and 100 mmol/L respectively, and the leaf phenotypes were recorded for 5 days, and thus the treatment time of 8 h and 10 mmol/L were selected as the treatment procedure promoted leaf senescence obviously. This information was added in Material and Method part in the revised MS.

  1. The connection between the observed effects and the underlying biochemical pathways could be explored more thoroughly. Include a discussion on how these findings align or contrast with existing literature.

Response: Thanks for your suggestion. Based on your suggestions we have made improvements in the discussion section. Please see the revision in our resubmitted manuscript.

  1. The conclusion appropriately summarizes the findings but lacks depth in terms of future research directions. Elaborating on how this research could pave the way for new studies or applications in agriculture would be valuable.

Response: Thank you for your valuable suggestions, which we have added to the discussion section of the article on the significance of studying this topic and our outlook for the future. We have added the following :“Controlling senescence can greatly improve crop yields and other plant characteristics, such as extending shelf life, especially in light of potential future food shortages and the use of plants as bioenergy sources [32]. Understanding how leaf senescence is regulated will help control senescence in the future, either through genetic modification or manipulation of key environmental triggers [33]. In the present work, we found that H2O2 could act a signal to promote leaf senescence by inhibiting H2S production. Thus, genetic manipulation on H2O2 metabolism-related genes such SOD, CAT and POD may provide a valuable tool to delay leaf senescence in crop or horticulture plants, thereby improving the yield. Please see the revision on line 301-309.

  1. In the References section, some recent studies could be added to provide a more current context to the research. Ensure that all references are up to date and relevant.

Response: Thank you very much for your advice. Based on your suggestions, we have updated and checked the references. Please see the revision in our resubmitted manuscript.

Round 2

Reviewer 1 Report

Comments and Suggestions for Authors

The authors have addressed the reviewer's comments in the revised manuscript. I appreciate these efforts.